# A Low Frequency Mechanical Transmitter Based on Magnetoelectric Heterostructures Operated at Their Resonance Frequency

**DOI:** 10.3390/s19040853

**Published:** 2019-02-19

**Authors:** Junran Xu, Chung Ming Leung, Xin Zhuang, Jiefang Li, Shubhendu Bhardwaj, John Volakis, Dwight Viehland

**Affiliations:** 1Materials Science and Engineering, Virginia Tech, Blacksburg, VA 24061, USA; cmleung@vt.edu (C.M.L.); xin2@vt.edu (X.Z.); jili4@vt.edu (J.L.); dviehlan@vt.edu (D.V.); 2Florida International University, Miami, FL 33199, USA; sbhardwa@fiu.edu (S.B.); jvolakis@fiu.edu (J.V.)

**Keywords:** electromechanical resonance frequency, magneto-elasto-electric coupling, transmitter–receiver system, kilohertz, magnetic fields, piezoelectric, magnetostrictive, ME sensors

## Abstract

Magneto-elasto-electric (ME) coupling heterostructures, consisting of piezoelectric layers bonded to magnetostrictive ones, provide for a new class of electromagnetic emitter materials on which a portable (area ~ 16 cm^2^) very low frequency (VLF) transmitter technology could be developed. The proposed ME transmitter functions as follows: (a) a piezoelectric layer is first driven by alternating current AC electric voltage at its electromechanical resonance (EMR) frequency, (b) subsequently, this EMR excites the magnetostrictive layers, giving rise to magnetization change, (c) in turn, the magnetization oscillations result in oscillating magnetic fields. By Maxwell’s equations, a corresponding electric field, is also generated, leading to electromagnetic field propagation. Our hybrid piezoelectric-magnetostrictive transformer can take an input electric voltage that may include modulation-signal over a carrier frequency and transmit via oscillating magnetic field or flux change. The prototype measurements reveal a magnetic dipole like near field, demonstrating its transmission capabilities. Furthermore, the developed prototype showed a 10^4^ times higher efficiency over a small-circular loop of the same area, exhibiting its superiority over the class of traditional small antennas.

## 1. Introduction

Magnetoelectric (ME) materials have the ability to convert energy between electrical and magnetic forms. Application of a magnetic field (H) results in a voltage output, or conversely an applied electric field (E) results in a magnetic flux change [1,2]. Consequently, ME materials have been investigated for potential applications as magnetic sensors, gyrator power converters, and field tunable communication devices. The ME effect was first found in Cr_2_O_3_ about 60 years ago, but the ME coupling effects in single-phase materials are very weak [1,2]. However, two-phase composites consisting of piezoelectric and magnetostrictive materials have been developed, which have very strong ME effects [2]. Although the two phases individually do not have an ME effect, bonded together they have an ME product tensor property. 

Our research team has previously developed ME heterostructures for passive (battery operated) magnetic sensors/receivers [3,4,5,6,7]. Their operation used external magnetic fields to excite magnetostrictive layers into mechanical vibration. In turn, this mechanical vibration was transferred to the bonded piezoelectric layer, resulting in a voltage output. Wang et al. [8] fabricated a longitudinal-longitudinal (LL) multi-push-pull structure using a piezoelectric layer bonded between two interdigitated electrodes (ID), and subsequently bonded to magnetostrictive layers forming a sandwich-like structure. A schematic of this ME sensor and its operation principal is depicted in Figure 1. Magnetic sensors based on these laminates and a charge amplifier detection method have been shown to have an extremely low equivalent magnetic noise floor of 5.1 pT/√Hz at 1 Hz. A large ME voltage coefficient of 52 Vcm^−1^Oe^−1^ has been obtained under optimized direct current DC magnetic bias for Metglas/Pb(Mg_1/3_Nb_2/3_)O_3_–PbTiO_3_ (PMN-PT) trilayer laminates [8,9].

Furthermore, large gains in the ME properties of ME laminates have been reported near the electromechanical resonance (EMR) frequency. The highest resonance ME voltage coefficient reported is 1100 Vcm^−1^Oe^−1^, based on the structure shown in Figure 1 [8,10,11]. Later, even higher ME voltage coefficients have been achieved by optimizations of geometric configurations and improvements of interfacial bonding between piezoelectric and magnetostrictive phases. For example, Li et al. reported a spin-coat/vacuum-bag method, which could improve interfacial bonding and increase ME coupling [12]. Furthermore, theoretical analysis has revealed that the resonance ME coupling coefficient in ME laminates can be significantly increased due to the improvement of the mechanical quality factor Q_m_ [13]. Based on the discussions above, Chu et al. proposed a composite with (1-1) connectivity, which exhibited an enhanced resonance ME coupling coefficient of 7000 Vcm^−1^Oe^−1^ [14]. This demonstrates the capabilities of ME sensors to have an excellent ability to receive low frequency (~30 kHz) and low power propagating signals, via utilizing this large ME gain effect at the EMR.

More recently, we have examined the transduction capabilities of ME heterostructures for power conversion devices and motors [15,16]. A highly efficient solid-state gyrator based on tri-layer composites consisting of two magnetostrictive ferrite layers epoxied to a Pb[Zr_(x)_Ti_(1−x)_]O_3_ (PZT) piezoelectric core placed between the ferrite layers was developed [17]. This electrical element used an input electric field to generate an output magnetic flux [16,18]. In turn, this flux induced current that flowed into an N-turns coil. Near the EMR, this ME gyrator exhibited a power conversion and transfer efficiency to a coil in the near field at about 90%, under low power density (0.61 mW/cm^3^) and 75% under a higher one of 1.83 W/cm^3^ [19]. These results signify that ME laminates do not have significant radio frequency (RF)-losses at very low frequencies (VLFs), and thus have a potential for higher transmission efficiency. Accordingly, ME laminate offers the potential to design an ME based transmitter–receiver system, when operating near the EMR. Such resonance-based transmitter–receiver would also have a DC bias tunable frequency.

A VLF transmitter is located in Cutler, ME, USA that transmits 2 MW of power [20]. It operates near 25 kHz and is capable of transmitting a signal around the world using a very small bandwidth at low data rates. The transmitter facility employs large towers spread across 2000 acres that has a radiation efficiency of 75% [20]. Portable receivers for this VLF system are magnetic sensors that are widely available. The transmitters for VLF electromagnetic waves with portability are needed [21]. Therefore, the reduction of the transmitter size needs to be investigated.

In this work, we are going to present the magnetic-field transmitting capabilities of the ME resonance sensors. The measurements of magnetic-field detection are presented, and we compared the proposed ME transmitter with a same-sized small current loop. Furthermore, the ME transmitter structure is also optimized to enhance the transmission.

## 2. Transmitter Construction and Operation Principal

### 2.1. Design and Operation

A multiferroic antenna structure consisting of piezoelectric and magnetostrictive materials has been proposed, which was used to create electromagnetic waves [22]. Depending on the concept, we investigated the transmitters based on ME laminates worked on at their EMR. The composite laminates consisted of a piezoelectric (PZT-5A) core and magnetostrictive layers on top and bottom of the core, as shown in Figure 2a. The PZT core consisting of five PZT fibers was attached with interdigitated (ID) electrodes to provide electrical actuation. According to the piezo properties, the longitudinal vibrations in the piezoelectric layers are transferred to the magnetostrictive layers due to mechanical (or strain)-coupling between PZT core and the Metglas layers at EMR. The magnetostrictive layers (e.g., Metglas) have the distinct property of converting mechanical vibrations to magnetic fields. Notably, these fields could be steady (static) magnetic fields, but upon radio frequency actuation, these fields are time varying in nature and thus will exhibit propagation. This described operation principle is also referred to as the electro-mechanical-magnetic effect.

### 2.2. Fabrication

To demonstrate the basic operation of the transmitter, three samples with varying construction were prototyped. The samples of the constructed transmitters are shown in Figure 2b. The first transmitter (of dimension 80 mm × 20 mm × 0.4 mm) was a Metglas/PZT/Metglas trilayer with six Metglas layers (three on either side). The PZT layer was poled in this sample. Poling refers to the orienting of all material dipoles in one direction by application of a strong electric field. Upon switching off the field, these dipoles do not return to their original position. Poling is necessary for inducing the required piezoelectric effect in the PZT layer. The second transmitter (dimension: 80 mm × 20 mm × 0.4 mm) was also a geometrically similar trilayer with a total of six Metglas layers, but the PZT layer was not poled. If the transmitter with the unpoled PZT layers was driven under a set input power, then the electro-mechanical effect should be negligibly low. The third transmitter (dimension: 53 mm × 20 mm × 0.3 mm) was a poled PZT layer, without any attached Metglas layers. If this transmitter was driven under a set input power, then there should only be electrical (non-magnetic) interference generated. Figure 2a, bottom shows the pictures of the copper coil with a diameter of 16 mm and length of 15.4 mm used as the receiver. There are about 200 turns of the coil.

## 3. Basic Transmitter Tests and Optimizations

### 3.1. Verification of the Electro-Mechanical-Magnetic Effect

First, we needed to confirm that the magnetic field around EMR detected by a receiver was indeed emitted by the ME transmitter through the electro-mechanical-magnetic effect in the laminate. Specifically, since unpoled PZT based ME laminated layer as well as the poled PZT layer without Metglas layers could not generate a magnetic-field, these could be used as a control sample. Therefore, an experiment was designed using the three different transmitters, as depicted in Figure 2c. To verify the electro-mechanical-magnetic effect, magnetic fields were measured by exciting the three samples with input signals at their resonance frequencies. Receiver search coil was placed at 40 cm along the longitudinal direction of the transmitter (Figure 2c).

The three fabricated samples were replaced at the transmitter side one by one and the magnetic field was measured. The measurement and excitation of the samples were conducted using the set-up shown in Figure 3. The voltage output received from the search coil was divided by the transfer function of the coil to obtain the detected magnetic flux density. The voltage output of the search coil was obtained using a preamplifier (SRS SR560, Stanford Research Systems, Sunnyvale, CA, USA) and a spectrum analyzer (SRS SR785, Stanford Research Systems, Sunnyvale, CA, USA), and it could also be observed on an oscilloscope (MSOX3014T, KEYSIGHT, Santa Rosa, CA, USA). In each case, the applied input power was gradually varied from 10 mW to 100 mW, while a resonance frequency, (i.e., 28 kHz) was used for the excitation. The input power was generated from a signal generator (AFG320, Tektronix, Inc., Beaverton, OR, USA), and amplified by an amplifier with 100 amplification. Only sample 1 (the poled PZT layer with Metglas layers) has an “electromagnetic” resonant frequency (around 28 kHz). Sample 2 (the unpoled PZT layer with Metglas layers) is unpoled, so it doesn’t have “electromagnetic” resonant frequency in low frequency regime. Although Sample 3 (the poled PZT layer without Metglas layers) has an “electroelastic” resonant frequency (around 35 kHz), it is not caused by the electromagnetic effect. Figure 4 shows the magnetic flux density detected by the receiver coil as a function of the input power for the three transmitters. The values measured using the trilayer with poled PZT as a transmitter were about three orders of magnitude higher than the ones from the poled PZT layer only. This verifies the assumption that the receiver is measuring the signal from sample 1 and not the electromagnetic interference (EMI) in the room. Furthermore, sample 2 (with unpoled PZT layer) was not polarized, and it still exhibits very weak electro-mechanical coupling, which could generate a tiny magnetic field by the magnetostrictive layers (Metglas). However, sample 3 (poled PZT layer without Metglas layers) could not generate any magnetic field without the magnetostrictive layers. This is the reason why there was still a small difference between the red curve and blue curve in Figure 4. Thus, for this case, the detected magnetic field from sample 3 is simply the background EMI in the room. Clearly, the magnetic flux densities detected by the receiver were generated by the ME transmitter with poled PZT layers.

The experiment above was then repeated at the EMR by using the trilayer structure with a poled PZT core and rotating it to 45° and 90°. The distance between the transmitter and receiver was maintained at 40 cm. As opposed to a 16 nT field observed with a 0° case, 9.6 nT and 0.82 nT field values were observed for 45° and 90°, respectively. Almost two orders of decrease in the field along the transverse direction (90° case) compares with the longitudinal direction (0° case). Thus, these experiments serve to verify the ME laminate operation.

### 3.2. Effect of Number of Metglas Layers

Next, we determined that the magnetic flux density generated by the ME transmitter could be enhanced by increasing the number of the Metglas layers. This determination was made by constructing ME transmitters with a varying number of bonded Metglas layers and measuring their ME coupling efficiency. The receiver was implemented using a standard Helmholtz coil with 50 turns on each side (radius: 45 mm). In this set-up, the ME laminate acted as a transmitter and the Helmholtz coil as a receiver. The input power at the EMR applied to the transmitter and the voltage outputs from the receiver were measured by an oscilloscope (KEYSIGHT MSOX3014T). The power efficiency (PE), defined as
(1)$$\mathrm{PE}\;=\frac{\mathrm{Ouput}\;\mathrm{power}\;\mathrm{from}\;\mathrm{the}\;\mathrm{receiver}}{\mathrm{Input}\;\mathrm{power}\;\mathrm{to}\;\mathrm{the}\;\mathrm{transmitter}},$$
is plotted for varying numbers of Metglas layers in Figure 5. 

The input power of the transmitter was calculated by the voltage, the current, and the phase between the voltage and the current. For the output power of the receiver, we used an optimized resistance connected to the receiver. Therefore, the output power was calculated by the voltage, the current and the phase difference on the optimized resistance [23]. The output power was measured from the receiver (i.e., a Helmholtz coil). Low power (4 mW–18 mW) measurements were conducted by fixing the input voltage to 10 Vrms. The power efficiency shown on the oscilloscope was then adjusted to its maximum value by varying the frequency.

Furthermore, the applied DC-magnetic field bias (Hdc) was varied from 0 Oe to 45 Oe in each case (Figure 5). Thus, the figure shows the results of the maximum PE at the EMR for ME laminates with different numbers of Metglas foils as a function of Hdc. The highest value of PE = 68.4% was found at Hdc = 30 Oe for an ME laminate with 20 Metglas layers. The dimension of this laminate was 80 mm × 20 mm × 1 mm. We will use this ME laminate with 20 Metglas layers (20M-trilayer) for the following studies, as it had the highest PE.

## 4. Transmitted Magnetic Fields, Efficiency and Pattern Measurements

Next, we investigate the transmission and efficiency capabilities of the ME laminate (a 20M-trilayer) at the EMR. These measurements allow us to understand the possibility of the communication properties of the ME transmitter, but, beyond that, by comparing a known small loop antenna, the transmission efficiency can also be estimated. We use this approach in our investigation. Finally, the magnetic field patterns of the proposed ME transmitter are also presented.

We use the set-up used in Section 3.1 for characterization of the ME transmitter (20M trilayer structure). We apply an input power 100 mW to the ME laminate at resonance frequency f_r_ = 28.17 kHz. The distance between the ME transmitter and the receiver, r, was then sequentially increased from 0.4 m to 1.35 m. Figure 6 shows the flux density detected by the receiver as a function of r. We measured the flux varying with distance in two directions, i.e., θ = 0° and 90°. We measured the dominant flux components along each direction. Specifically, B_r_ (along θ = 0°) and B_θ_ (along θ = 90°) were measured. As shown, the magnetic-flux levels varied between 30 nT to 1 nT in the provided distance range. We will understand more about these values by comparing our results with a small current-loop antenna in the following subsection.

### 4.1. Comparison with a Small Current-Loop Antenna

To understand the transmission capabilities of the presented ME transmitter, it is apt to compare them with a canonical antenna structure. The size of the proposed current loop is as same as the ME transmitter, which is 16 cm^2^. As is well known, a horizontal small circular loop is equivalent to a vertical magnetic small-dipole erected through the loop’s center. Given that and the fact that the ME transmitter has a magnetic-dipole like resonance along the longitudinal direction, this comparison is appropriate (Figure 6, right).

First, using derivations provided in Appendix A, the near field of the small loop of radius a can be derived as
(2)$$\left|B_{r}\right|\approx \frac{\mu_{o}}{\sqrt{2}}\sqrt{\frac{3}{{\pi \eta k}^{4}}\frac{Z_{o}R_{r}P_{\mathrm{in}}}{{\left(R_{r}{+Z}_{o}\right)}^{2}}}\left(\frac{2\cos\theta }{r^{3}}\right)$$
and
(3)$$\left|B_{\theta }\right|\;\approx \;\frac{\mu_{o}}{\sqrt{2}}\sqrt{\frac{3}{{\pi \eta k}^{4}}\frac{Z_{o}R_{r}P_{\mathrm{in}}}{{\left(R_{r}{+Z}_{o}\right)}^{2}}}\left(\frac{\sin\theta }{r^{3}}\right).$$

Here, P_in_ is the input power applied at the port of the loop antenna, Z_o_ (= 50 Ω) is the impedance of the input transmission line, R_r_ ≈ 31,171$\frac{S^{2}}{\lambda^{4}}$, with S = πa^2^ is the radiation resistance of the antenna, η = 377 Ω is the free-space radiation impedance and permeability of the free-space is given by μ_o_ = 4π × 10^−7^ H/m. k = 2π/λ is the propagation constant for a wavelength λ. To match the measurement conditions, P_in_ = 100 mW was chosen for the loop antenna. r is distance of the observation point from the center of the antenna and θ is the angle as shown in the right figure of Figure 6. For a reasonable comparison, we chose the loop area to be S = 16 cm^2^ (same as the area of ME transmitter 8 × 2 cm^2^). Note that a factor of 1/√2 is added in B_r_ and B_θ_ to obtain root mean square (RMS) fields for the loop antenna for a rational comparison with the measurements. Measured B_coil_ is essentially RMS flux density as obtained from the oscilloscope voltage (refer to Figure 3).

Figure 6 reveals a close match between the field-distance profiles from the loop antenna and the measured ME transmitter. Notably, according to Labels (2) and (3), ratio |B_r_|/|B_θ_| = 2 is expected. On average, this ratio was found to be ≈ 1.84 in the experiments. This is a reasonable agreement since we expect diffractions, reflection and refractions within the room, including that from the ground. Note that, because of low frequency and near-static fields (λ = 10 km), these interferences manifest themselves as constant additions of field, instead of inducing oscillations as generally expected in such distance profiles in the microwave regime. Furthermore, the empirical curve fitting suggests that the measured magnetic flux density decayed as 1/r^2.6^. This is comparable with theoretical 1/r^3^ given various sources of errors in the measurements. Finally, we also observe a difference in the field levels from a loop antenna and the ME transmitter. This is further discussed in the efficiency subsection next.

We also measured the magnetic flux density generated by the ME transmitter as a function of the distance with the highest power (500 mW) that we could apply. In Figure 7, the black curve was measured with 100 mW, and the red one was measured with 500 mW. Extrapolating the 500 mW fitting curve, it can be estimated that a 1 fT flux could be detected around 200 meters. The background magnetic noise floor in an open environment at EMR (~30 kHz) is about a few hundred fT/√Hz [24]. Therefore, a higher input power level or a greater number of transmitters could be added to increase transmitted magnetic fields. 

### 4.2. Transmission Efficiency of the Magnetic Field

Efficiency calculations generally require knowledge of total transmitted power, which requires knowledge of 3D far-fields. However, this could be difficult for the kHz range antenna, due to large wavelength. Most modern antenna chambers support measurements only above a few tens of MHz.

Under these restrictions, we infer the transmission efficiency of the proposed ME transmitter by comparing its performance with the theoretical current-loop discussed above. We note that field ratio |B_r_|/|B_θ_| = 1.84 confirms a similarity of ME transmitter with the loop antenna. Furthermore, we know that the ME transmitter at the EMR works like a magnetic dipole along the longitudinal direction due to the magnetic moment’s direction in the magnetostrictive layer. It is a reasonable assumption that the field-profile from ME transmitter and current-loop should match (as further confirmed in the pattern measurements in the next section) across the 3D space.

We also notice that the magnetic field produced by the loop is two orders smaller than the ME transmitter. For example, B_r_ at 90 cm from the ME transmitter is 2.9 nT and from the loop is 0.039 nT. This information, along with known radiation efficiency of the loop antenna, can be used to calculate the radiation efficiency of the proposed antenna. We know that the radiation efficiency is proportional to magnetuc flux squared. Thus, the efficiency ratio of ME antenna (η_ME_) and circular loop (η_loop_) will follow as
(4)$$\frac{\eta_{\mathrm{ME}}}{\eta_{\mathrm{loop}}}={\left(\frac{\left|B_{r_{\mathrm{ME}}}\right|}{\left|B_{r_{\mathrm{loop}}}\right|}\right)}^{2}\approx 5.5{\times 10}^{3}.$$

Thus, the radiation efficiency of the proposed ME transmitter is expected to be three to four orders higher than a loop antenna of same area. This confirms the superior operation of the proposed ME transmitter in near-field and far-field. The proposed transmitter is efficient especially for the cases when the packing area is small, since it can provide higher efficiency within the same size.

We note that the efficiency of the loop antenna is extremely low (η_loop_ = 1.2 × 10^−19^) due to a ≪ λ restriction at kHz range [25]. That is, since the antenna is non-resonant, its radiation resistance is negligible as compared to the impedances of the typical transmission line. This means that most power will simply reflect back from the antenna port. In other words, the factor 1-|S_11_|^2^ is negligible. This is a typical challenge for antennas that are operating at much longer wavelength than their size. Even with such severe impedance matching challenges, the proposed ME transmitter provides substantial improvements in the low frequency (~30 kHz) transmission, causing the radiation efficiencies to be estimated in the order of 10^−16^, which is three orders larger than for the loop antenna of the same size.

### 4.3. Pattern Measurements

To specify the orientation of the ME transmitter, we assume that the ME transmitter’s longitudinal direction along the *Z*-axis, the width along the *X*-axis, and the thickness along the *Y*-axis. The magnetic field pattern measurements were done by sweeping the elevation angle θ in the XZ (or 001)-plane, YZ (or 010)-plane and XY (or 100)-plane. We chose to measure B_r_ component for these measurements by orienting the axis of the search coil towards the center of the sweeping circle. The radius of the circle was 0.5 m. The input power applied was 100 mW. The ME transmitter was working at its EMR for all the cases. The measurement set-up was similar to Figure 3, but applied for varying θ.

The normalized magnetic field pattern profiles are shown in Figure 8a–c, respectively. The maximum value of the magnetic flux density was measured in Figure 8d of the red curve, which was 4.75 nT. The fields in XZ- and YZ-planes are normalized to their maximum value, as both fields were found to be in the tens on nT range. For the XY-plane, the fields are found to be one order smaller than XZ, YZ-planes. The normalization is conducted with a maximum of the XZ-plane to show the NULL along the XY-plane (Figure 8c).

The B_r_ field component follows the cosine field profile for XZ-(001) and YZ-(010) planes, confirming a magnetic-dipole (or horizontal small-loop) like operation predicted by Label (2). We note that the detected field in the XY (100)-plane in Figure 8c was one order smaller than Figure 8a,b, which effectively represents a NULL at θ = 90° based on Label (2). However, the power in the (100) plane was still measurable, which is probably due to a slight asymmetry of the ME transmitter and receiver.

Additionally, comparison of the normalized magnetic-field patterns measured using a single ME laminate under input powers of 100 mW and 500 mW was made along the XZ-(001) plane. Both patterns are shown in Figure 8d. Their overlap confirms the linear operation of the ME transmitter, i.e., the transmitted fields linearly increase with the applied input power. This is an important observation, given the nonlinear physics of the ME laminates in consideration.

## 5. Conclusions

In this work, we consider the optimization of an ME transmitter and measured its magnetic fields at EMR (~30 kHz). As opposed to large current radiation antennas for kHz, the proposed approach used mechanical vibrations to excite electromagnetic radiation using piezo and magneto laminates. Furthermore, the conversion efficiency was optimized by increasing the number of Metglas layers. The enhanced magnetic flux density provides proof-of-concept for miniaturization of low frequency transmitters based on mechanical-magnetic effects.

Our results clearly demonstrate small portable devices with significantly enhanced magnetic flux density in the 30 kHz frequency range. Even with small efficiency, the presented values are several orders higher than a current-loop antenna of the same size. Furthermore, the background magnetic noise floor in an open environment at EMR (~30 kHz) is about a few hundred fT/√Hz [25]. A higher input power level or a larger number of transmitters could be added to increase transmitted magnetic fields, but this will be at some compromise to the size. These effects along with impedance matching concerns are areas of future work in this direction.

## Figures and Tables

**Figure 1 sensors-19-00853-f001:**
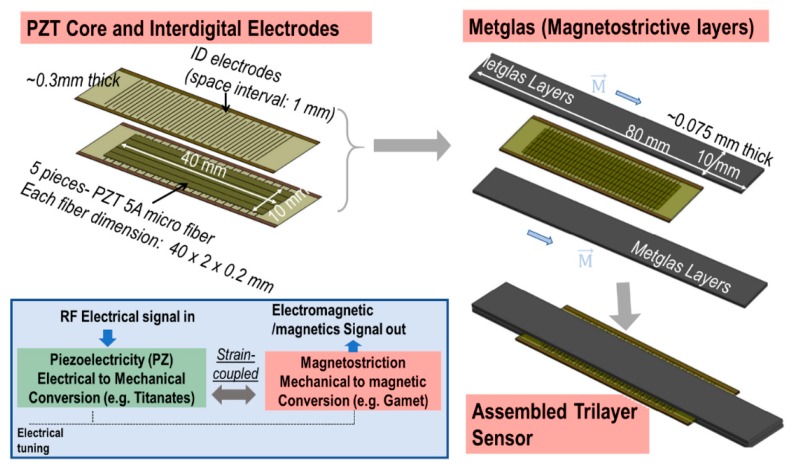
Schematic of the proposed ME transmitter, its construction and operation principal using piezo and magneto layers.

**Figure 2 sensors-19-00853-f002:**
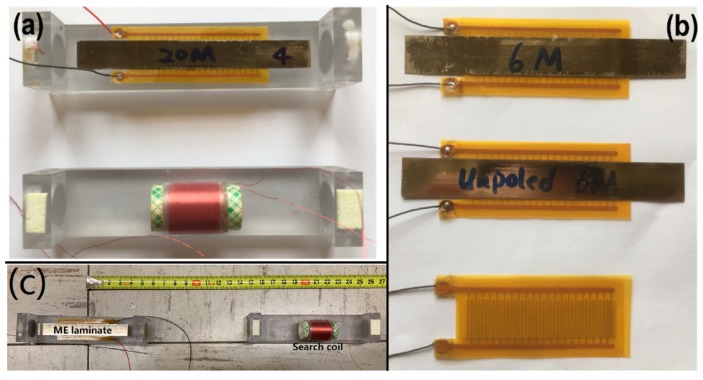
(**a**) fabricated ME transmitter and search coil used for test and measurements, (**b**) three samples- 6 Metglas layers (6M) based ME transmitter, unpoled 6M ME transmitter and poled Piezo layers used in characterization, and (**c**) experimental set-up for field measurements.

**Figure 3 sensors-19-00853-f003:**
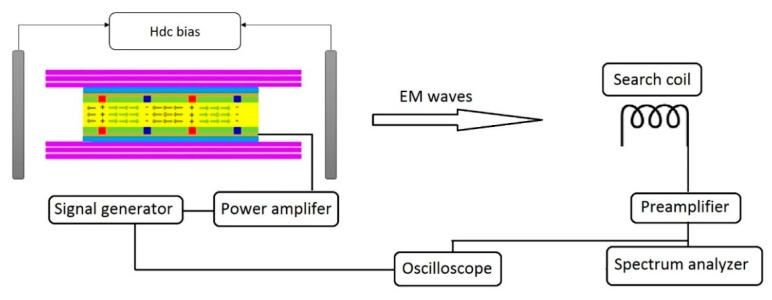
Measurement set-up and excitation method used for the characterization of the ME transmitter.

**Figure 4 sensors-19-00853-f004:**
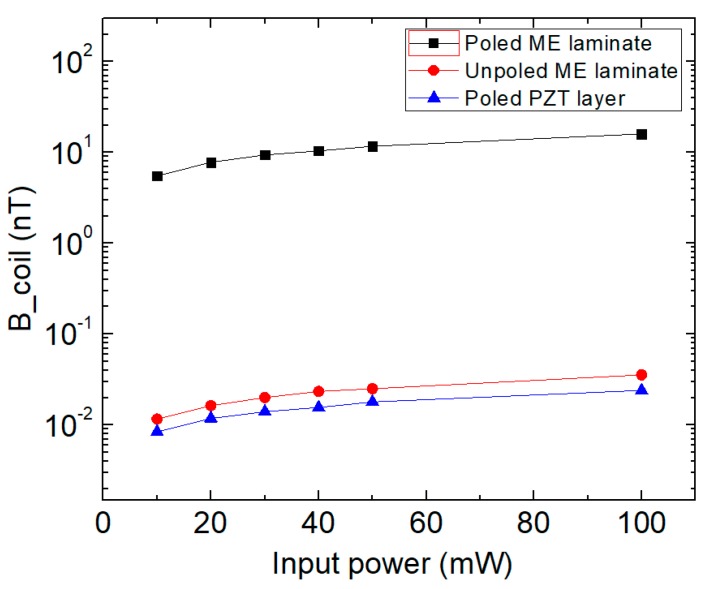
The magnetic flux densities detected by the receiver as a function of the increasing input power to the different transmitters.

**Figure 5 sensors-19-00853-f005:**
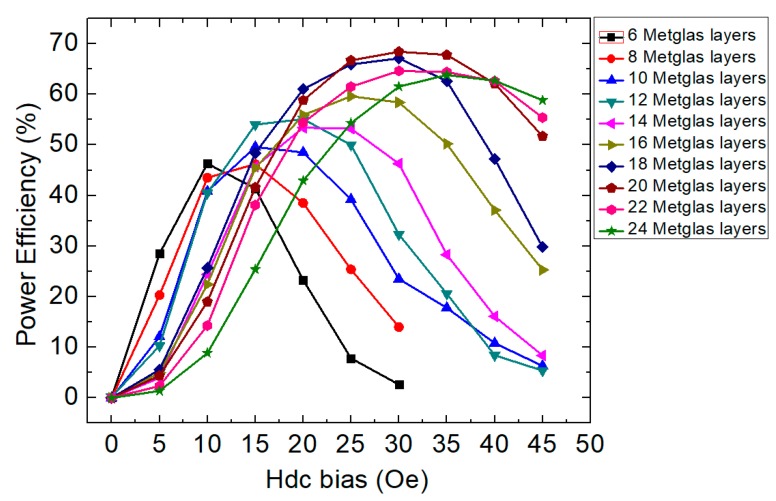
Power efficiency of the transmitter–receiver system as a function of the increasing magnetic DC bias using the ME laminates with the different numbers of Metglas layers.

**Figure 6 sensors-19-00853-f006:**
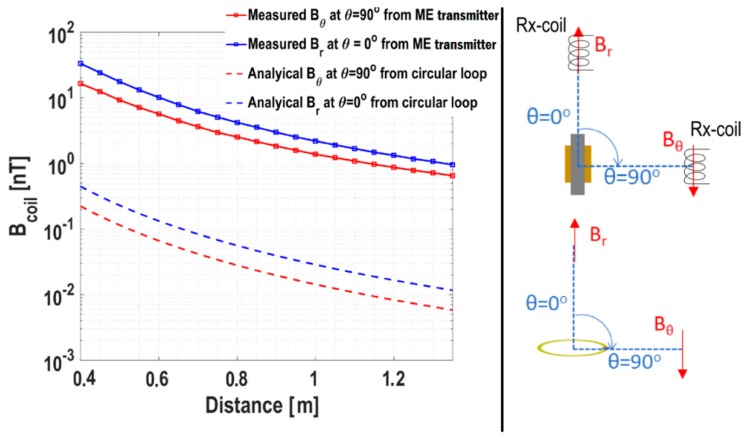
**Left figure**: the magnetic flux density detected by the receiver (Rx-coil) as a function of the distance along longitudinal (θ = 0°) and transverse (θ = 90°). **Right figure**: the configurations of the ME transmitter and the circular loop. The results are compared with analytical results for a small current-loop antenna.

**Figure 7 sensors-19-00853-f007:**
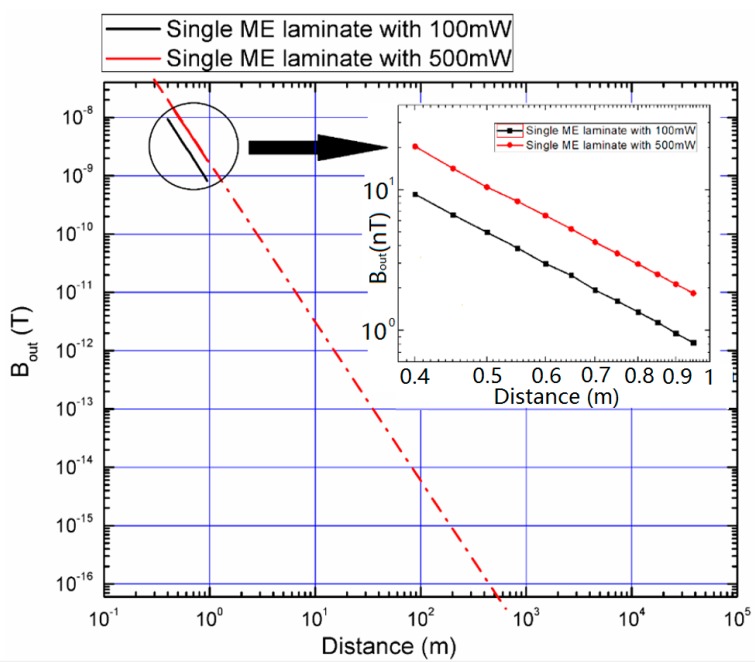
The magnetic flux density (near-field) generated from the ME transmitter as a function of the distance between the ME transmitter and the receiver. The dash dot line is the extended line on the curve measured using a single ME laminate with 500 mW.

**Figure 8 sensors-19-00853-f008:**
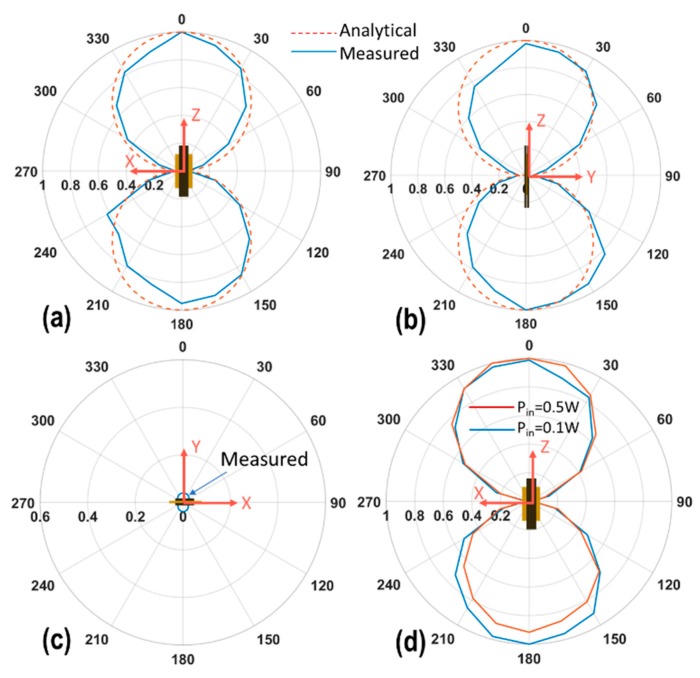
The measured normalized radial magnetic field (B_r_ (θ)) of the ME transmitter as a function of elevation angle θ in (**a**) XZ (or 001)-plane, (**b**) YZ (or 010)-plane, (**c**) XY (or 100)-plane, and (**d**) linearity performance of the ME transmitter as shown with two different power levels.

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
