# Peer review of "A Low Frequency Mechanical Transmitter Based on Magnetoelectric Heterostructures Operated at Their Resonance Frequency"

_sensors, 2019, doi:10.3390/s19040853_

Round 1
Author Response
Thank you so much!
Reviewer 2 Report
This paper concerns a new class of electromagnetic emitter materials, based on magnetoelectric (ME) composite material working at its electromechanical resonance frequency in objective to produce the propagation of an electromagnetic wave.
The paper present the design and operation principle of the heterostructure, basic transmitter tests and an optimization (mainly the number of Metglas layers) of the ME composite, a comparison with a small current-loop antenna to achieve the efficiency of the ME antenna and ends with pattern measurements.
The paper is well written, but some sentences need to be improved.
Major remarks:
In the first paragraph of section 2.1, what do you mean by "the dynamic and propagating electromagnetic fields are anticipated and measured" and in particular "are anticipated "? It is not clear.
What is the PZT material used (reference, property)? Figures 2 suggest that they are MFC.
How do you achieve the polarization of the PZT layer? Similarly, for the second sample how do you obtain and verify that the PZT layer is unpoled?
What is the number of turns of the search coil (figure 2)?
Give more informations on devices of the measurement set-up.
It is indicated that since sample 2 is not polarized it has no low frequency resonance in low frequency regime. Has this been verified? If so, what is the value of the first resonance frequency?
It is indicated for sample 2 (with unpoled PZT layer) that it cannot generate signals and the detected magnetic-field is simply the background EMI in the room, which would explain the associated curve in Figure 4. This explanation is not necessarily the right one. Tests should be performed with another sample composed of non-polarized PZT and layers of ferromagnetic material (with the same dimensions as Metglas) with a much lower magnetic susceptibility than Metglas to be able to verify it. Even if the PZT layer is not polarized, an electro-mechanical coupling (via electrostriction) is present. The signal emitted is therefore not only due to electromagnetic noise, but also to very low ME coupling. This would also explain the difference between the curves unpoled ME laminate and the poled PZT layer. What is your opinion? Have you considered this point?
For the optimization of the number of Metglas layers, why did you use a Helmholtz coil? What configuration did you use (distance between transmitter and receiver)? Give more informations on the Helmholtz coil used (number of turns).
Regarding the first two sentences on page 6 (section3.2, lines 175-176) it is not clear what is being done. Is the resistance added to the Helmholtz coil?
In optimizing the number of layers, do all layers have the same dimensions for all configurations (including layer thickness) or does the thickness vary between configurations?
What are the EMR frequencies obtained for each configuration?
For the results of figure 5, is the input power used the same regardless of the frequency of use?
Give informations on the small current-loop antenna.
For section 4, in the comparison is the input power the same?
The comparison is performed by choosing a current loop with the same surface area as the ME transmitter. Is this the right comparison? Why not choose a current loop producing a field identical to the ME at a given point? The design of the loop could be done from the analytical formulas.
How you define the efficiency in relation (4)? For the ME antenna and the loop antenna.
In analysis of the pattern results, it is indicated the linear operation of the ME transmitter and this is noted as important information given the non-linear behaviour of ME composites. Okay, but are the field levels involved enough to excite the non-linearities. Was the magnetic flux density in the ME laminate measured?
Minor remarks:
P1, keywords: “magneto-elasto-electric coupling” instead of “magneto-elasto-electric”.
P1, l35: It seems that references other than the reference [1] indicated are more appropriate in view of what is indicated.
P1, l41: “In turn, this mechanical vibration…” instead of “In turn, this mechanical force…”.
P3: For the first paragraph of section 2.1 there's a little redundancy with the introduction (lines 96 to 99).
P3, l79: the reference indicated ([1]) is not the correct one.
,P3 l95: “with interdigitated (ID) electrodes…” instead of “with interdigital (ID) electrodes…”.
P3: Figure 3.c is not very readable.
P4, l129: “was designed using the three…” instead of “was designed using three…”.
P4, l149: The acronym EMI is not defined.
P8: figure 7, the y-axis for the zoom is not readable.
P10, l313: Define the coefficient kr.
P10: Define the coefficient h in relation (9).
P10, l314: “of the radiator from the…” instead of “of the radiator form the…”.
P11: Complete the references [20] and [21].
P12: Reference 25 is not accessible.
Author Response
Thank you for your comments.
Please take a look at the attachment.

Reviewer 3 Report
Please, read the attached file.

Author Response

(The authors gave the same response as above.)
